# Self-Healing Structural Materials

**DOI:** 10.3390/polym13142297

**Published:** 2021-07-13

**Authors:** Seongpil An, Sam S. Yoon, Min Wook Lee

**Affiliations:** 1SKKU Advanced Institute of Nanotechnology (SAINT) and Department of Nano Engineering, Sungkyunkwan University (SKKU), Suwon 16419, Korea; esan@skku.edu; 2School of Mechanical Engineering, Korea University, Seoul 02841, Korea; 3Institute of Advanced Composite Materials, Korea Institute of Science and Technology, Chudong-ro, Bongdong-eub, Jeonbuk 55324, Korea

**Keywords:** self-healing, self-repair, structural composites, mechanical properties

## Abstract

Self-healing materials have been developed since the 1990s and are currently used in various applications. Their performance in extreme environments and their mechanical properties have become a topic of research interest. Herein, we discuss cutting-edge self-healing technologies for hard materials and their expected healing processes. The progress that has been made, including advances in and applications of novel self-healing fiber-reinforced plastic composites, concrete, and metal materials is summarized. This perspective focuses on research at the frontier of self-healing structural materials.

## 1. Introduction

Nature-inspired self-repairing strategies have been explored in biomimetic engineering with the aim of restoring damaged materials. A variety of types of infrastructure must be protected as human activity expands from being terrestrial-based into such environments as submarine and space. Various self-healing methods have recently been developed and tested. These bioinspired engineered materials, i.e., materials that “self-heal” after external damage, have been studied since the early 1990s [1,2]. The damage to engineering materials is mostly repaired via the process of systematic transport and the polymerization of healing materials in the damaged area. For this reason, the majority of early studies on self-healing materials were conducted using soft materials [2]. Polymeric materials were found to be more manageable when encapsulated in microcapsules, prepared as nanofibers, or used in a reversible form, such as a thermoplastic or supramolecular materials; in addition, these materials were viable and did not require external stimulation to initiate the healing process [3,4]. However, from the perspective of mechanical strength, healing materials are inevitably soft and weak. To date, numerous self-healing materials have been reported; however, most of these are mechanically soft and weak and are thus not appropriate for practical applications involving structural materials. Therefore, the next goal in the development of self-healing materials is to devise materials with sufficient mechanical strength for use as structural materials. For example, the walls of a nuclear power plant or an airplane frame must possess sufficient strength under extreme conditions, including long-term fatigue conditions. Thus, healing materials for use in these applications are also required to be strong, meaning that traditional polymer-based materials are inherently unsuitable. Moreover, large structures are difficult and expensive to maintain, and detecting and repairing defects is challenging from a technical perspective. Solutions that are realistic from both technical and material points of view are required to identify and repair structural problems that may arise in thick reactor walls or fuselages of aircrafts during flight. To this end, new approaches have been investigated to repair hard materials, such as concrete composites [5,6,7,8,9], carbon fiber-reinforced plastic (CFRP) composites [10,11,12], steel, and aluminum [13,14,15,16]. Owing to the wide range of applications for self-healing materials, novel strategies and new materials are being actively developed. The challenges that have been overcome to integrate these self-healing materials into full-fledged structures on real construction sites are highlighted in this review. Furthermore, the effects of the various self-healing techniques on panel motion under load and the techniques used to heal the cracks that develop are discussed.

This study surveys cutting-edge self-healing approaches, with a particular focus on structural materials. There is an adage that states “a small crack breaks a big dam.” High-rise buildings and long bridges are susceptible to accidents and disasters every hour of every day; therefore, the focus of self-healing technology must necessarily shift toward such large structures. In the past, the main purpose behind the use of self-healing materials was to repair scratches and prevent rust; however, the research community concerned with self-healing materials should now focus on preventing accidents caused by the collapse of bridges or the rupture of aircraft structures. Once motivated by the desire to repair physical damage, self-healing technology is predicted to be used in future to safeguard the world’s infrastructure and protect lives.

## 2. Self-Healing in Polymers and Their Composites

Fiber-reinforced composites are attracting increasing attention because of the demand for lightweight structures and materials for the construction of buildings and other large structures that require high strength [17]. In particular, the increasing use of high-strength and lightweight composite materials composed of glass fiber, aramid fiber, polyethylene (PE) fiber, and carbon fiber in large-scale structures has resulted in an increase in the cost-efficiency. The global market for these materials, the use of which has expanded rapidly over the past few decades, was valued at $90 billion in 2019 but is expected to grow to $113.6 billion by 2024 as a result of the composites incorporated into sporting equipment, automobiles, and protective equipment [18]. However, items susceptible to long-term use, such as mechanical and thermal fatigue, remain a major challenge. In particular, major accidents involving commercial airliners and nuclear plants have been reported to be induced by small cracks [19,20]. Aircraft structural health monitoring and maintenance costs to prevent massive casualties or economic losses are expected to reach $5.5 billion by 2025 [21]. The lack of technical evaluation methods for inspecting, monitoring, and detecting damage to large structures remains an important shortcoming that needs to be resolved. To this end, many studies have been conducted to fabricate structural composites with self-healing capabilities. Self-healing polymeric materials based on elastomeric materials, thermoplastics/thermosets, ionomers, or supramolecular materials have been widely studied for their anticorrosive and fatigue repair properties [22,23,24,25,26]. Using a number of chemical crosslinking processes, the mechanochemical reaction (or network formation) of polymeric materials aims to achieve selective and repeatable mending. It is highly required to be autonomous healing but is also accelerated by the activation energy induced by thermal and photochemical stimulation. These types of polymers are easy to handle and are compatible with well-known commercial products, such as PDMS and urethane, thus being attractive from practical and economic points of view. Moreover, the required mechanical properties of polymer materials are high strength and stiffness. To this end, polymeric self-healing materials are expected to play a very important role in self-healing systems.

In the first study on the repair of CFRP composites, healing materials were encapsulated in microcapsules to prevent delamination failure of the CFRP composites [27]. Dicyclopentadiene (an healing agent) capsules (d = 166 μm) mixed with epoxy resin were ruptured and released into the crack plane (Figure 1a). The interlaminar fracture toughness of a double-cantilever beam specimen at room temperature and 80 °C was restored to 40% and 80% of its initial value, respectively. Another approach to repair delamination damage involved the use of a thermoplastic polymer as the matrix material [28]. Polyurethane in combination with the Diels–Alder reaction was found to achieve repeated healing of the delamination with healing efficiencies of 85% and 75% for the first two cycles. The damaged specimens were compressed and heated for 2 h to 135 °C, 90 °C, and 70 °C. This was the first method to accomplish the intrinsic healing of the CFRP matrix material (Figure 1b); however, the mechanical strength of the matrix material could not meet the standard required of structural materials. D’Elia et al. [29] mimicked brick-and-mortar structures in glass and poly(borosiloxane) composites (Figure 1c). Stiff blocks and supramolecular polymers imparted mechanical strength and repeatable healing ability to the material, resulting in a structural composite model with high strength and good thermoplastic properties. Successful healing that enabled the structure to fully recover its original strength at the MPa level was repeatedly demonstrated at room temperature.

Thus far, the self-healing function has been provided by polymers in composite matrix materials [31,32]. Until recently, soft healing materials could not be used to repair hard materials. However, a new approach, which does not require the synthesis of new healing substances, has been developed to solve the delamination problem experienced by carbon composite materials. A stitching technique aided by Joule heating was first applied to carbon-fiber-reinforced thermoplastic (CFRTP) composites. Carbon fabric laminates were stitched with electrically conductive threads, which were then used for heat conduction to achieve voltage-induced Joule healing. As the temperature of the conductive threads increased, the thermoplastic matrix became fluidic above its melting point and highly bendable. This approach constitutes a simple but versatile method for the postforming of a carbon composite while maintaining its solid state and hardness [30]. This approach can also be used to repair flaws in FRP composites. Despite the superior tensile strength of the reinforcing fibers, the interfacial adhesion was relatively weak. Notably, this method can be applied to FRPs without the need for a soft material as the healing agent. This is because the matrix material only undergoes a phase change for several minutes, during which time voids in the matrix can be filled immediately and repeatedly. The voids within the composite can be eliminated by melting and cooling the matrix, i.e., by switching the voltage that is applied to the thread on and off, respectively. This feature is useful for repairing tiny flaws that are difficult to detect but that could potentially lead to serious failures.

## 3. Self-Healing in Cement-Based Materials

Cement is one of the most widely used materials in the construction of plants for civil engineering projects and in architectural structures [33]. Cement is a brittle material that can easily crack when subjected to vibration and tension. Within a certain period after a structure has been built, the structural performance of cement begins to deteriorate, which is manifested by the appearance of cracks. Cracks allow harmful air, moisture, and chemical components to penetrate the structure from the outside, further degrading its performance. In other words, once a crack occurs inside the cement, the damage spreads rapidly from the crack to the starting point. Additional cracks may occur because of the corrosion of the reinforcing bars inside the structure as a result of moisture and chloride ions penetrating the cement material; alternatively, the structure may collapse because of degraded performance due to, for example, a corrosion-induced decrease in the cross-sectional area of the reinforcing bars. Therefore, increasing the lifespan of cement structures and preventing further damage require microcracks to be repaired quickly. Because of the nature of building structures, cracks internal to these structures are difficult to repair by repairing only the corresponding parts. Self-healing of cement-based materials is not an entirely new problem. In 1836, self-healing in water-retaining structures was reported by the French Academy of Sciences; the hydration of cement-paste-like calcium carbonate or polyacrylic hydrogel was utilized for repairing cracks in cementitious materials [34,35,36,37,38]. This approach has now been extended to treating cracks with a size from a few microns to the millimeter scale, thus achieving a more practical use in real structures. In particular, this methodology is applied to large-scale infrastructures, such as buildings, tunnels, dams, or bridges, and the lifespan and durability of such constructions are of high interest worldwide. From the maintenance point of view, cracking is one of the main causes of deterioration and damage in concrete structures. Concrete is renowned for its maintenance and repair costs ($147/m^3^), which are about twice the production costs ($60–80/m^3^). In the United States, approximately $18 to 21 billion is spent on the maintenance, repair, and replacement of damaged structures. Considering the structural properties of concrete materials, it is still a big challenge to find effective healing agents and encapsulation techniques; however, significant progress has been made in this direction. Conventionally, cracks in structures are repaired using filling materials, such as epoxy and mortar, via the outer walls of the structures. In one example, a capsule containing a self-healing material that was crosslinked by exposure to ultraviolet rays was applied to a concrete surface [39,40]; however, the results achieved with this method depend on the penetration limit of ultraviolet rays. The conventional crack repair method is effective only for limited crack healing on the surface of the structure; the filling of internal cracks not exposed to the exterior is difficult. In addition, a method that enables microcracks to undergo self-healing using microcapsules has recently been proposed [41,42]; however, broken microcapsules always exist in externally or internally damaged areas. An additional problem is that the damage can be repaired only once.

Sodium silicate is a representative healing material for concrete, and the sodium silicate approach, which involves mixing microcapsules containing sodium silicate with cement, has been investigated [43,44]. The capsules were 300–700 μm in size with a wall thickness of 5–20 μm. This approach is simple and effective; the microcapsules were dispersed in the breaks in the concrete and released the healing compound at the locations where the cracks occurred (Figure 2a) [45]. Biomineralization using various axenic/nonaxenic microorganisms can enhance the self-healing of concrete materials. Ersan et al. quantified the crack closure and water impermeability performance using bioconcrete [46]. This green technology is promising as it has been shown to increase the service life of cementitious structures to as long as 94 years [47]. The precipitation of calcium carbonate (CaCO_3_) around bacterial clusters is the main healing mechanism in this approach. The precipitated minerals grow to connect crack openings of 10^3^ μm to 1 mm; however, the process takes 14–28 days to complete (Figure 2b). Another bacterial healing method using flow networks was developed at Bath University [45]. The spores of Bacillus pseudofirmus DSM 8715 were contained in a 500 mm layer in the middle of the control-mix concrete. A vascular flow network formed by polyurethane tubes provided an efficient feeding system for the healing material in scaled-up construction applications. A trial test was conducted, comprising the in situ construction of concrete panels using a shape-memory polymer (SMP)-based system for crack closure [9]. A two-dimensional network of channels with a diameter of 4 mm was created, allowing the networks to pass in front of the SMP tendons (Figure 2c). Notably, the network approach is more beneficial than the aforementioned approach based on microcapsules. This is because the empty volume of the ruptured capsules adversely affects the mechanical strength of the entire material, whereas the vascular channel network can supply the healing agent repeatedly. Long-term and sustainable self-healing capacity is especially required in structural applications, as the increased durability of structures not only enhances the safety but also lowers the cost of large-scale infrastructure in many countries [48]. The appropriate dispersion of conductive fillers (e.g., carbon nanotubes, graphene, polyacrylonitrile-based carbon fibers, pitch-based carbon fibers, or graphite) in a composite of cement may increase its electrical conductivity to exceed that of general cement. Thus, heat generation due to electric pressure becomes possible. A polymer filler that can be melted (e.g., poly(ethylene oxide) (PEO), PE, polyurethane, polycarbonate, or polycaprolactone) can be formed inside the cement composite (Figure 2d). This approach enables high temperatures (e.g., in the range of 40–80 °C) to be generated inside the cement composite due to electric pressure; when a crack occurs, the polymer filler melts, and the crack can be repaired by the polymer, which fills the inner microcracks [49]. As previously described, the use of self-healing methods to repair the cement composite, including conductive fillers and polymer fillers, can prevent further fracture of the cement composite, extending its lifespan. General cement was found to be rapidly destroyed after a fatigue test of approximately 1000–2000 cycles. Thus, when general cement is used, the lifespan of the structure may be very short. The prepared cement composite was heated after a fatigue test of approximately 1000 cycles (e.g., heat was generated by applying electricity) to melt the incorporated PEO powder. Rapid failure was delayed, even in fatigue tests of 8000 cycles or more. Even in the event of the formation of internal cracks, in the case of the cement composite, heat generation is induced via the conductive filler, and the internal cracks can be effectively repaired as a result of being filled with polymer melted by the generated heat.

## 4. Self-Healing in Metals and Metal Matrix Composites

As is well known, metal is the strongest and most widely used material from early human history to the present day. It is hard to envisage the occurrence of healing in such a hard material; this phenomenon has been studied in a number of experiments and computational works. The number of publications on self-healing in metals is clearly lower than the number of reports on other materials, but the interest has been rapidly increasing in recent years [50,51]. In comparison with the fast and efficient reactions in polymers, the atomic bonding is strong in metals; thus, a high energy is required to transport the healing agent to the cracks. The capability of self-healing in metals lies in this basic concept for various methods, including impregnated capsules, alloy atoms, coatings, or electro-heating. The precipitation of supersaturated solute atoms at high temperature is most intensively studied in this concept. For this reason, the temperature should be sufficiently high to promote the lattice diffusion of the solutes but not too high. In short, the possibility of attaining crack or void healing in metals is closely related to the atomic mobility of the solute atoms. A complete recovery of the mechanical properties using this approach has not been achieved yet; however, the key factor to enable the restoration of a crack lies in the mobility of the solute atoms, which can fill the crack. The mobility of such atoms is promoted at an elevated temperature, such as half the value of the melting temperature. In this concept, ferrite and aluminum alloys induced by gold, tungsten, molybdenum, boron, and nitride have been used [52,53,54]. It should be noted that the mechanical properties of entire materials have not been evaluated yet. The most important reason behind the necessity to heal a structure is preventing its sudden failure. Strong metal materials are required to recover their mechanical strength not only by filling cavities but also via chemical bonding. The following approaches to achieve self-healing metals are not autonomic; they require an external energy source (e.g., high temperature) and cannot be used to recover the original mechanical properties; thus, the meaning of “self-healing” becomes somewhat loose. Regardless of which one of the aforementioned methods is employed, i.e., soldering cubes/capsules or coating agents, it should be noted that the incompatibility between the healing agent and the healed material may cause other problems, such as lattice mismatch and alloy-composition changes. Nonetheless, it is meaningful to discuss recent achievements on cavity healing in metals.

One possible approach to heal the cavity in austenite stainless steels was demonstrated experimentally by Shinya et al. [55]. They reported the precipitation of boron nitride (BN) on creep cavity surfaces during creep and its beneficial effect on creep rupture properties. It was suggested that the precipitation of BN on the creep cavity surface endows austenitic stainless steel with self-healing capabilities for creep cavitation with an associated increase in the creep rupture strength and ductility. As shown in Figure 3, boron and nitrogen atoms aggregate on the cavity surface, thus suppressing cavity growth. Two basic mechanisms are here considered the healing process: (1) the interstitial diffusion of healing materials (B, N) and (2) the precipitation of the atoms on the cavity surface. Importantly, the healing efficiency may depend on the mobility of healing materials, which can be accelerated under high pressure or temperature.

In general, this type of diffusion and precipitation is performed at a temperature (T) of 0.40–0.65 T_m_ (where T_m_ is the melting temperature) to promote the activation of diffusant atoms [57]. Fang et al. studied Fe–Au alloys that induced gold-rich precipitates at a temperature of 550 °C (T_m_ = 1064 °C) for the autonomous filling of cavities. The cavities and precipitates in the model alloy were analyzed quantitatively via synchrotron X-ray tomography. The gold precipitates were found at the cavity surfaces, causing a change in the crack morphology. The model predictions were in good agreement with the experimental results [58]. Similar studies were performed for Fe–Mo alloys at temperatures of 540–565 °C for the potential self-healing alloy model. It was found that multiple creep cavities were filled by Fe_2_Mo precipitates at grain boundaries (see Figure 4) [59]. The precipitate size was on the scale of several micrometers, which means that this is a practical and effective strategy to fill real cracks in metals. At higher temperature, more numerous and denser precipitates fill the creep cavities. From the aforementioned studies, it is expected that the supersaturated solute materials in the alloy can provide an opportunity to heal metal materials and further extend their creep lifetime.

It is clear that the heated metal loses its structural strength under a temperature as high as T/T_m_ ≈ 0.5. Moreover, it must be pointed out that the physical properties may not be fully recovered with a healing material weaker than the original metal, and the composition of the alloy may change during the heating process, resulting in a failure to achieve the designed properties. These problems are crucial and have to be addressed in the near future.

## 5. Self-Healing in Ceramics and Ceramic Composites

Ceramic is another traditional and promising material for use at high temperature and under corrosive conditions. It has been developed for composite materials to improve their brittleness and crack resistance; these include alumina (Al_2_O_3_), mullite (3Al_2_O_3_·2SiO_2_ or 2Al_2_O_3_·SiO_2_), and silicon carbide (SiC). Excellent strength properties and heat resistance are required in internal combustion engines as they are subjected to loads of 500–800 N/mm^2^ and temperatures of 1000 °C [60]. However, only a limited number of studies have focused on this aspect in the last two decades. In the early 2000s, self-healing structural ceramics was realized via the oxidation of nonoxide healing agents at high temperature [61,62,63,64,65]. The recovery of the mechanical properties (static strength, fatigue strength, and fracture toughness) of a precracked (100–200 μm) mullite/SiC or Al_2_O_3_/SiC composite was investigated at an elevated temperature of 900–1400 °C. The heating process was performed for 1–300 hr in air, and nanosized SiC particles were used as the healing agent. It has been reported that a smaller particle size favors oxidation at low temperature, but a particle size below 10 nm is not sufficient to fill the cracks. The oxygen delivery in ceramic materials is usually low; therefore, it takes a relatively long time for the oxidation process to occur (10^2^–10^3^ hr). Furthermore, 0.2 vol.% MnO was designed as an activator in Al_2_O_3_/30 vol.% SiC composites to promote oxygen delivery to the healing agent (see Figure 5) [66]. According to Osada et al., the production of oxides to fill the crack was accelerated; the required heating temperature was then dropped, and the time for complete healing was reduced by a factor of 6000 to 10 min. This new idea inspired by the bone healing process had a significant contribution toward rendering the self-healing of brittle ceramic materials more realistic.

## 6. Concluding Remarks

Over the past 30 years, new materials and successful approaches that expand the field of self-healing applications have been explored. In the next few decades, the performance of self-healing materials under extreme conditions must be considered. A viable self-healing technology must be developed to resolve safety issues that inevitably arise for large structures. FRP composites, concrete, and metals, which are representative building materials, are more important in this respect. We should devote more attention to endowing these hard materials with the ability to undergo self-healing, which was considered impossible until now. Thus, self-healing technologies make it possible to contribute to the protection of lives and properties by using new materials and groundbreaking approaches.

## Figures and Tables

**Figure 1 polymers-13-02297-f001:**
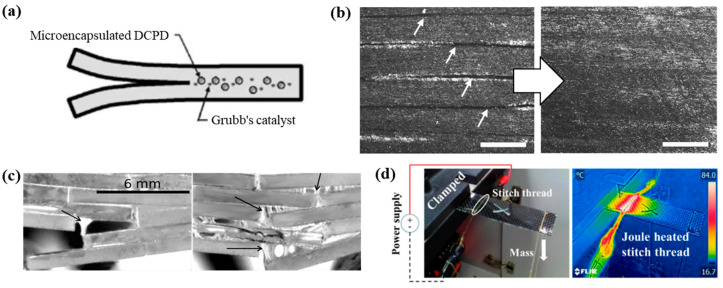
(**a**) Self-healing specimen in which microcapsules of the healing agent and the catalyst are embedded in the polymer matrix, and healing is autonomic. Reprinted with permission from [27]. (**b**) Optical microscopy images of a composite laminate before and after healing. The scalebar is 50 μm. Reprinted with permission from [28]. (**c**) Optical micrographs of brick-and-mortar structure composites under a bending test. The displacement rates are 1 and 0.1 mm/min. Reprinted with permission from [29]. (**d**) Snapshot and infrared image of a CFRTP specimen stitched with an electrically conductive thread for Joule heating. Reprinted with permission from [30].

**Figure 2 polymers-13-02297-f002:**
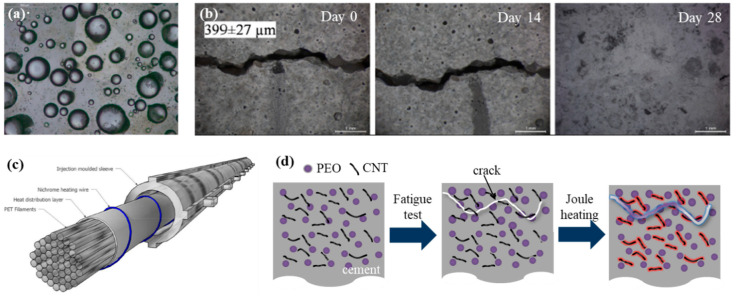
(**a**) Microscope image of the microcapsules composed by an emulsion (54-wt% mineral oil, 42-wt% sodium silicate, and 4-wt% emulsifier). Reprinted with permission from [45]. (**b**) Evolution of a crack at the specimen surface containing biomortar. Crack widths are given in the micrograph. Reprinted with permission from [46]. (**c**) Schematic of the concrete crack closure system of a shape-memory PET tendon. Reprinted with permission from [9]. (**d**) Schematic of the self-healing cement composite induced by Joule heating. Reprinted with permission from [49].

**Figure 3 polymers-13-02297-f003:**
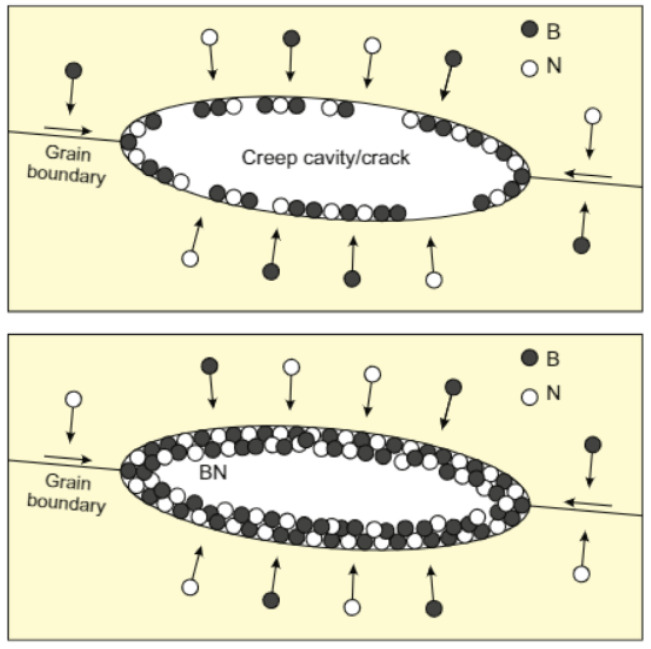
Illustration of BN precipitation on the creep cavity surface in stainless steel. Reprinted with permission from [56].

**Figure 4 polymers-13-02297-f004:**
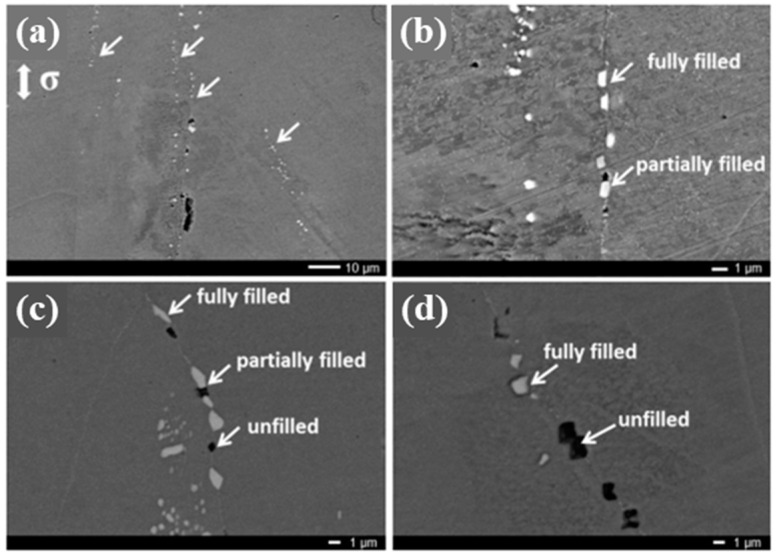
Micrographs of the Fe–Mo alloy after creep under a stress of 160 MPa at a temperature of 565 °C demonstrating cavities and precipitation at grain boundaries parallel to the loading direction at selected locations. Reprinted with permission from [59].

**Figure 5 polymers-13-02297-f005:**
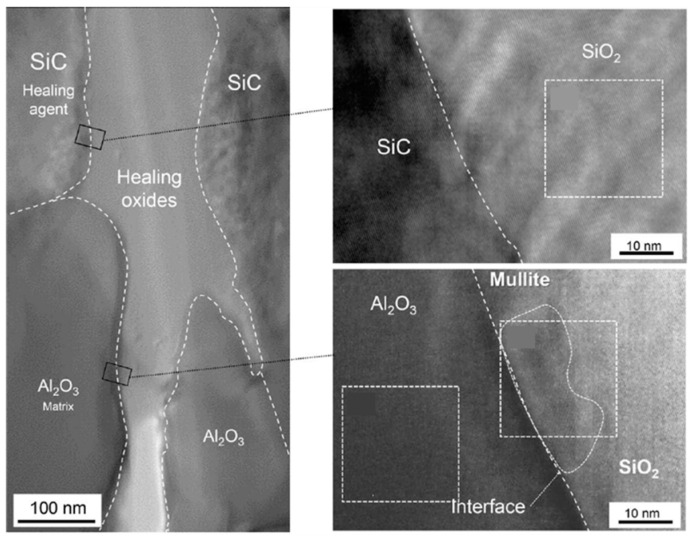
Structure of the gaps in Al_2_O_3_/SiC composites healed at 1473 K. High-resolution transmission electron microscopy of healed cracks at the interface between SiC and SiO_2_ and between Al_2_O_3_ and SiO_2_. Reprinted with permission from [66].

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
