# Peer review of "Self-Healing Structural Materials"

_polymers, 2021, doi:10.3390/polym13142297_

Round 1

Reviewer 1 Report

The manuscript entitled "Self-Healing Structural Materials" presents a critical vision of the current state of the art of the technologies available for obtaining hard self-healing materials. The work also presents the latest advances in the three most important types of hard materials: fiber-reinforced plastic composites, cements and metallic materials. This perspective is very well written and easy to follow and understand. In addition, the references are very up-to-date, which gives an idea of the current relevance of this topic.

For all these reasons, I consider that the manuscript can be accepted in the present form.

Author Response

We thank the Reviewer for the comments and suggestions for our revised manuscript. 

Reviewer 2 Report

This manuscript must be considered as a review article. Therefore, this manuscript needs essential modifications as follows:

  • "A review article" must be presented in the title of the manuscript.
  • The English language should be improved throughout the manuscript. Please write the manuscript in reporting style or using passive sentences (not active sentences, e.g., we, I, etc.), see Abstract, Line # 13 " Herein, we discuss cutting-edge self-healing technologies for hard materials". This comment applies to the whole manuscript.
  • Line # 13, "hard materials". Therefore, a review of self-healing in ceramic and ceramic composites should be included in this paper
  • The Title of Sec. 2 should be improved to be "Self-Healing in Polymers and their composites". In addition, this section should be extended, and the following Refs. Should be added:
    • Print ISBN: 9780128184509 eBook ISBN: 9780128184516
    • Print ISBN: 978-3-527-33439-1 ePDF ISBN: 978-3-527-67021-5
    • Print ISBN: 978-0-470-49712-8
    • ISBN: 978-3-527-31829-2
  • The Title of Sec. 3 should be improved to be "Self-Healing in Cement-Based Materials". In addition, this section should be extended, and the following Refs. Should be added:
    • DOI1007/978-94-007-6624-2
    • DOI1007/s12205-020-0090-6
    • DOI 1016/j.jobe.2021.102834
    • DOI 1016/j.jmrt.2020.04.053
    • DOI 1016/j.conbuildmat.2019.03.117
  • In-Line # 175, it should be corrected to be Sec."4."
  • In-Line # 223, the number of Sec. must be changed.
  • The title "Filling cavity in Metals" should be improved to be "Self-Healing in Metals and Metal Matrix Composites". In addition, this section should be extended, and the following Refs. Should be added:
    • DOI 1007/978-3-319-32778-5
    • DOI 1007/s40195-020-01102-3

Author Response

We thank the Reviewer for the comments and suggestions. Please find the attached file that include our responses to the Reviewer’s specific concerns.

Round 2

Reviewer 2 Report

The authors have successfully addressed all my comments.  Therefore, I recommend the publication of this manuscript.